# Impact and cost-effectiveness of potential interventions against infant respiratory syncytial virus (RSV) in 131 low-income and middle-income countries using a static cohort model

Ranju Baral [ID] , Deborah Higgins, Katie Regan, Clint Pecenka

► Prepublication history and supplemental material for this paper are available online. To view these files, please visit the journal online (http://dx.doi.org/10.1136/bmjopen-2020-046563).

Center for Vaccine Innovation and Access, Program for Appropriate Technology in Health, Seattle, Washington, USA

**Correspondence to**
Dr Ranju Baral; rbaral@path.org

## ABSTRACT

**Objectives** Interventions to prevent childhood respiratory syncytial virus (RSV) disease are limited and costly. New interventions are in advanced stages of development and could be available soon. This study aims to evaluate the potential impact and cost-effectiveness of two interventions to prevent childhood RSV—a maternal vaccine and a monoclonal antibody (mAb).

**Design** Using a static population-based cohort model, we evaluate impact and cost-effectiveness of RSV interventions, from a health systems perspective. The assumed baseline efficacy and duration of protection were higher for the mAb (60%–70% efficacy, protection 6 months) compared with the maternal vaccine (40%–60% efficacy, protection 3 months). Both interventions were evaluated at US$3 and US$5 per dose for Gavi and non-Gavi countries, respectively. A range of input values were considered to explore uncertainty.

**Settings** 131 low-income and middle-income countries.
**Participants** Pregnant women and live birth cohorts.
**Interventions** Maternal vaccine given to pregnant women and mAb given to young infants.
**Primary and secondary outcome measures** Disability-adjusted life years averted, severe case averted, deaths averted, incremental cost effectiveness ratios.
**Results** Under baseline assumptions, maternal vaccine and mAbs were projected to avert 25% and 55% of RSV-related deaths among infants younger than 6 months of age, respectively. The average incremental cost-effectiveness ratio per disability-adjusted life year averted was US$1342 (range US$800–US$1866) for maternal RSV vaccine and US$431 (range US$167–US$692) for mAbs. At a 50% gross domestic product per capita threshold, maternal vaccine and mAbs were cost-effective in 60 and 118 countries, respectively.
**Conclusions** Both interventions are projected to be impactful and cost-effective in many countries, a finding that would be enhanced if country-specific Gavi cofinancing to eligible countries were included. mAbs, with assumed higher efficacy and duration of protection, are expected to be more cost-effective than RSV maternal vaccines at similar prices. Final product characteristics will influence this finding.

## Strengths and limitations of this study

► This is one of the first studies to examine the potential impact and cost-effectiveness of maternal vaccines and monoclonal antibodies for respiratory syncytial virus (RSV) prevention, across 131 low-income and middle-income countries.
► This study compares products with uncertain characteristics using the latest available data on vaccine characteristics, supplemented by the target product profile to inform the model parameters.
► A range of input values were considered to explore uncertainty, insights from which are useful to inform subsequent intervention development.
► Final product characteristics and product prices will determine the relative cost-effectiveness of RSV interventions.

## INTRODUCTION

Respiratory syncytial virus (RSV) is a common cause of acute lower respiratory illness (ALRI) among children younger than age 5, causing between 41 000 and 118 000 child deaths per year globally.[1 2] RSV disease is most severe among young infants, and the burden is highest in low-income and middle-income countries (LMICs), where more than 99% of RSV deaths occur.[2] Emerging evidence indicates the unrecognised burden of RSV among children in low-resource settings is also significant, with up to 10% of young infant deaths attributable to RSV infection.[3 4]

Existing RSV interventions are limited and cost prohibitive, even in high-income countries.[5] Several prophylactic interventions are currently under development.[6 7] Multiple maternal vaccine candidates designed to protect against RSV illness in infants are in relatively advanced stages of development and expected to be available for global use in the coming years.[6] Monoclonal antibodies

(mAbs) are available and in use for high-risk babies in high-income countries. However, the available mAbs are not only expensive but require multiple doses during the RSV season. Long-lasting more affordable mAbs that are easier to deliver in low-resource settings are in advanced stages of development.[8] Given the extent of the global RSV disease burden—especially in low-income countries (LICs)—and the lack of efficacious and cost-effective therapeutic options, these new interventions are expected to be included in Gavi, the Vaccine Alliance's, portfolio,[9] subject to licensure, prequalification and cost characteristics.

In this paper, we estimate the potential impact and cost-effectiveness of a maternal vaccine and a mAb, both designed to avert RSV disease burden in young infants in LMICs. We compare each intervention against a scenario of no intervention and against each other. Results from this study illustrate the potential benefits of these products and will help inform decisions around further development. This analysis will also inform global and LMIC decision-makers likely to face choices about whether and which interventions to introduce.

## METHODS

We examined the potential impact and cost-effectiveness of a single-dose RSV maternal vaccine administered to pregnant women at 24–36 weeks gestation, and of a single-dose mAb given to infants directly at birth across 131 LMICs, compared with no intervention. Both interventions were evaluated independently using a static cohort model. For maternal vaccine, infants born 2 weeks following maternal immunisation were considered as protected to allow time for immune transfer from mother to children. All children receiving mAbs were considered protected immediately.

We examined the impact and cost of interventions from the health systems perspective over the period 2030–2039 (10 years), assuming nationwide introduction in 2030. Primary input values for a baseline scenario are given in table 1. Key model outputs include cases averted, severe cases averted, hospitalisations averted, deaths averted, disability-adjusted life years (DALYs) averted and the incremental cost per DALY averted due to RSV interventions. Given the lack of country-specific cost-effectiveness thresholds across LMICs, we used a willingness-to-pay threshold of 0.5 times the gross domestic product (GDP) per capita in each country.[10] Results are summarised by WHO regions, World Bank income group, and Gavi eligibility to understand impact by country group. All monetary units are adjusted to 2016 US dollars.

### Disease burden

Disease burden inputs including disease incidence, severe disease incidence, incidence of hospitalisations and mortality were derived from a comprehensive systematic review paper.[2] We combined country-specific disease incidence estimates in children under 5 years of age with a representative developing-country estimate to generate incidence by granular age band in each country. To generate incidence of severe disease, hospitalisation and hospital mortality, we used developing-country estimates.[2] Estimated hospital deaths were adjusted by multiplying by 1.98 to account for community deaths and influenza coinfection.[2] The actual values of disease burden inputs are given in table 1.

Some RSV interventions under development have shown promising results in their ability to avert all-cause lower respiratory tract infections (LRTIs) among children,[11] in addition to RSV infection. Thus, we also explored the potential impact of both RSV interventions on all-cause LRTI, based on emerging burden data, using estimates from the Global Burden of Disease Study 2017,[12] and assuming a uniform distribution of disease among children 1–12 months of age. Further, we assumed 11.5% of all-cause LRTI cases would result in severe cases[13] and 40% of all severe cases would result in hospitalisation.

### Intervention introduction and coverage

The leading RSV intervention candidates could be available for use in the next 5–8 years.[7] We assumed both interventions would be available by 2030 and all countries would begin national introduction in 2030.

All pregnant women attending antenatal care (ANC) visits were assumed eligible to receive RSV maternal vaccine. To project the number of pregnant women per country, we added country-specific stillbirths[14] to the United Nations Population Division annual birth projections.[15] We estimated maternal vaccine coverage during the 24-week to 36-week vaccination window by examining country-specific ANC first-visit timing,[16] country-specific ANC coverage[17] and the WHO's recommended ANC timing based on the focused ANC model guideline.[18] Details on methods used in estimating maternal vaccine coverage during ANC within a specific gestation window is described elsewhere.[19]

All live births were considered eligible for mAbs. All infants were assumed to be covered at the BCG vaccine birth dose coverage levels adjusted to the timeliness of vaccine receipt. Country's overall BCG coverage were derived from the most recent WHO/UNICEF estimates of national immunisation coverage.[20] Timeliness of BCG birth dose receipt was derived using the methods described in the literature.[21 22]

Coverage levels for both interventions for each country are projected to improve by 3 percentage points each year until coverage reaches 70%, and after that by 1 percentage point each year until reaching 95% coverage. This projection was made to correspond with methods applied during the Gavi vaccine investment strategy.[9]

### Intervention characteristics

Our analysis assumed a single-dose maternal RSV vaccine would be given to pregnant women between 24 and 36 months of gestation, inferred based on the WHO preferred product characteristics (PPCs)[23] and

**Table 1** Key input parameter values used for modelling

| Input | RSV maternal vaccine | RSV mAb | Sources |
|---|---|---|---|
| **Intervention-specific inputs** | | | |
| Target population | 126 million (annual average number of pregnant women, between 2030 and 2039, across 131 countries) | 124 million (annual average live births between 2030 and 2039, across 131 countries) | Birth estimates and population growth rate;[15] stillbirth rates[14] |
| Intervention schedule | Single-dose vaccine given during weeks 24–36 of gestation, as a part of ANC | Single-dose mAb given to newborn at birth | WHO[23] and ClinicalTrials.gov;[24] expert opinion |
| Efficacy against RSV endpoints | Baseline: cases=40.9%; hospitalisation=41.7%; death=59.6% Minimum scenario: 30% (for all endpoints) Maximum scenario: 90% (for all endpoints) | Baseline: cases=60%; hospitalisation=60%; death=70% Minimum scenario: 30% (for all endpoints) Maximum scenario: 90% (for all endpoints) | Novavax, Inc[11] and WHO,[23 25] expert opinion |
| Duration of protection against RSV* | Baseline: 3 months Minimum scenario: 4 months Maximum scenario: 6 months | Baseline: 6 months Minimum scenario: 4 months Maximum scenario: 6 months | WHO;[23 25] expert opinion |
| Efficacy against all-cause LRTI† | Cases=25%; hospitalisation=25%; death=39% | Cases=25%; hospitalisation=25%; death=39% | Novavax, Inc;[11] expert opinion |
| Duration of protection against all-cause LRTI† | 6 months | 6 months | Novavax, Inc;[11] expert opinion |
| Intervention coverage | Derived from ANC coverage (average 84%, range: 40%–96%, in 2030) | Derived from BCG coverage (average 82%, range: 48%–98%, in 2030) | Demographic and Health Surveys,[16] UNICEF[17] and WHO[18 20] |
| **Common to both interventions** | | | |
| Disease burden | | | |
| Incidence of RSV-ALRI | Country-specific incidence for 0–5 years for envelope (35.3–65.6). Developing-country estimate by narrow age band for case distribution by age. Annual incidence per 1000 children | | Shi *et al*[2] |
| | 0–27 days | 40.0 | |
| | 28 to <3 months | 45.7 | |
| | 3–5 months | 99.6 | |
| | 6–11 months | 98.8 | |
| | 12–23 months | 79.1 | |
| | *Rescaled to match country-specific incidence envelope* | | |
| Incidence of severe RSV-ALRI | Developing-country estimates with uniform age distribution | | Shi *et al*[2] |
| | Annual incidence of severe RSV-ALRI per 1000 children | | |
| | 0–5 months | 36.1 | |
| | 6–11 months | 24.7 | |
| | 0–59 months | 10.2 | |
| Hospital admissions for RSV-associated ALRI | Annual hospital admissions for RSV-associated ALRI per 1000 children | | Shi *et al*[2] |
| | 0–5 months | 20.2 | |
| | 6–11 months | 11 | |

**Table 1** Continued

| Input | RSV maternal vaccine | RSV mAb | Sources |
|---|---|---|---|
| Hospital case fatality | Hospital case fatality risk (%), by age group | | Shi et al[2] |
| | 0–5 months | 2.2 | |
| | 6–11 months | 2.4 | |
| RSV-ALRI mortality | Hospital deaths 2.2 (*adjusted for community deaths*) 0.9 (*adjusted for influenza activities*) | | Shi et al[2] |
| Incidence of all-cause LRTI | Country-specific; By ages: early neonates (0–7 days), post neonates (7–28 days), late neonates (1–12 months); burden for post neonates, uniformly distributed across ages by month | | IHME[12] |
| Incidence of severe LRTI | 11.5% of all incidence resulting in severe cases | | Assumed (based on the estimates used in Rudan et al[13]) |
| Hospital admissions for LRTI | 40% of all severe cases resulting in hospital admissions | | Assumed |
| Mortality due to LRTI | Country specific, early neonates, post neonates, late neonates; burden for post neonates uniformly distributed across ages by month | | IHME[12] |
| Age distribution of LRTI | Assumes uniform distribution of burden across months by age | | Assumed |
| **Costs** | | | |
| Intervention cost | US$3 per dose in Gavi countries; US$5 per dose in non-Gavi countries | | Assumed |
| Intervention delivery costs | Mean incremental economic cost of delivery per dose: US$0.63 in LICs; US$1.73 in LMICs and UMICs | | Immunization Costing Action Network[29] |
| Treatment cost | Cost of managing severe pneumonia in LMICs (outpatients US$53; inpatients US$250) | | Zhang et al[36] |
| Vaccine introduction dates | National introduction starting 2030 | | Product development timeline, assumed |
| **Other assumptions** | | | |
| DALY weights | Severe ALRI=0.21; non-severe ALRI=0.053 | | IHME[37] |
| Duration of illness | Severe ALRI=10 days; non-severe ALRI=5 days | | Graham and Anderson[38] |
| Length of hospital stay | Length of stay for severe pneumonia in LMICs, 6.4 days | | Zhang et al[36] |
| Healthcare seeking | Health seeking for children with pneumonia, country specific | | WHO[39] |

*Duration of protection in the minimum scenario is higher than in the baseline scenario. For maternal vaccine baseline, we assume duration of protection data from a recent clinical trial that failed to meet the primary endpoint. Nonetheless, in anticipation that a successful product would likely have higher duration of protection than 3 months, we evaluate the minimum scenario at 4 months duration of protection.
†Used in adjunct scenario only. The adjunct scenario evaluates intervention impact on all-cause LRTI mortality.
ALRI, acute lower respiratory illness; ANC, antenatal care; DALY, disability-adjusted life year; LICs, low-income countries; LMIC, low-income and middle-income country; LRTI, lower respiratory tract infection; mAb, monoclonal antibody; RSV, respiratory syncytial virus; UMIC, upper-middle-income country.;

other ongoing clinical trials.[24] We based vaccine efficacy and duration of protection on data from one of the first maternal vaccine candidate phase III clinical trials (table 1).[11] Other maternal vaccines are in clinical development which may have improved efficacy. Given the uncertainty in vaccine characteristics, scenario analyses included a range in efficacy (30% to 90%) and duration of protection afforded to infants (3–6 months).

Our analysis assumed a single-dose mAb would be given to newborns at birth, would have 60%–70% efficacy, and would offer protection for 6 months, inferred based on the PPCs[25] and other studies.[26 27] As with the maternal vaccine, we varied efficacy and duration of protection in scenario analysis. We assumed neither intervention contributed to herd immunity, and that efficacy did not wane during the period of protection.

### Intervention price and delivery costs

For both interventions, we assumed a per-dose price of US$3 in Gavi-eligible countries and per-dose price of US$5 in LMICs not eligible for Gavi support. Traditionally mAbs are more expensive to produce than a vaccine and will likely have higher market price than a vaccine. If Gavi decides to support RSV interventions once they

are available, Gavi-eligible countries would likely be able to access the interventions at varying prices depending on their transition status.[28] We refrained from projecting individual country Gavi eligibility or intervention prices due to significant uncertainty, and instead evaluated a range of intervention prices in sensitivity analyses.

Given the paucity of data on maternal immunisation and mAb delivery costs in LMICs, we used delivery cost estimates for other vaccines derived from the Immunization Costing Action Network repository.[29] Unit costs of delivering RSV interventions were US$0.63 for LICs and US$1.73 for LMICs. We accounted for vaccine/mAbs wastage at 5% and a buffer stock at 25% of demand in the introduction year, and at 25% of the incremental demand in subsequent years.

### Health service costs
Very few studies have analysed the cost of managing RSV in children, especially in LMICs.[30–35] Hospitalisation costs also vary widely. In Bangladesh, for example, hospitalisation averages US$74, whereas in China it averages US$662. Given limited RSV-specific information in LMICs, we used the average cost of treating pneumonia in young children, identified in a systematic review[36] as US$53.26 and US$250.04 per outpatient and inpatient episode, respectively. We assumed that severe cases seek inpatient care and non-severe cases seek outpatient care.

### Cost-effectiveness analysis
We calculated intervention costs by multiplying the number of doses (estimated number of pregnant women receiving vaccine for maternal vaccine and estimated number of live births for mAbs) with the unit cost of delivery and cost per dose. We estimated averted healthcare costs by multiplying the estimated number of non-severe/severe cases averted by the costs of an outpatient/inpatient episode.

Vaccine impact was calculated by multiplying the respective disease burden in children born 2 weeks after maternal vaccination with vaccine efficacy. The mAb impact was calculated by multiplying disease burden with the BCG coverage estimates and mAb efficacy. We estimated health outcomes including severe/non-severe cases averted, hospitalisations averted, deaths averted and DALYs averted for each country and year. Disability weights for non-severe and severe ALRI were used to compute DALYs.[37] Further, we assumed duration of illness at 5 days for non-severe disease and 10 days for severe disease.[38] The length of a hospital stay for severe disease was assumed to be 6.4 days.[36] Both undiscounted and discounted DALYs (at 3% discount rate) were generated for the analysis. We also accounted for variation in health-seeking practices by using healthcare use data from children younger than 5 years receiving pneumonia care.[39]

We calculated incremental cost-effectiveness ratios (ICERs) for each country by dividing the net cost of intervention by the net DALYs averted by the intervention.

### Sensitivity analysis
We conducted one-way sensitivity analysis by changing the values of key input parameters, including intervention efficacy, duration of protection, anticipated coverage and intervention price. Alternate scenarios that changed one or more input parameters to evaluate results sensitivity were also considered. In an adjunct scenario, we evaluated how different interventions show impact on all-cause LRTI mortality, using the efficacy and duration of protection values as suggested by recent clinical trial data,[11] and disease burden for all-cause LRTI from the 2017 Global Burden of Disease Study.[12]

### Patient and public involvement
Patients were not included in this modelling study.

## RESULTS
### Disease burden without interventions
Over the 10-year period, about 41.94 million non-severe cases, 15.28 million severe cases, 11.48 million hospitalisations and 504 963 deaths among children younger than 6 months of age in 131 LMICs are projected (table 2). Seventy-three Gavi-eligible countries accounted for 70% of the mortality burden. Most deaths would occur in sub-Saharan Africa (36%, 47 countries), followed by South Asia (26%, 8 countries).

### Expected health outcomes with intervention
RSV maternal vaccine, under the baseline scenario, has the potential to avert 2.97 million non-severe cases, 2.63 million severe cases, 2.03 million hospitalisations, 126 552 deaths and 3.73 million DALYs (discounted) among children younger than 6 months of age across all countries over 10 years (table 2). Globally, about 17% of severe RSV cases and 25% of RSV-related deaths among infants under 6 months of age would be averted by RSV maternal vaccine, which is roughly 13 deaths averted per 100 000 vaccinated pregnant women.

An RSV mAb, under the baseline scenario, is expected to avert 19.47 million cases of non-severe disease, 7.18 million severe cases, 5.40 million hospitalisations, 276 933 deaths and 8.19 million DALYs (discounted) among children younger than 6 months of age across all countries over 10 years (table 2). Globally, about 55% of RSV deaths among infants younger than 6 months of age would be averted with RSV mAbs—equivalent to approximately 28 averted deaths per 100 000 newborns receiving the intervention.

Under alternative scenarios using varying efficacy and duration of protection assumptions (minimum and maximum scenarios), the RSV maternal vaccine is estimated to avert between 84 934 and 356 346 deaths over 10 years; and the RSV mAb is expected to avert roughly 84 864 and 356 057 deaths. Assuming both interventions are able to affect all-cause LRTI, as suggested by recent clinical trial data,[11] either intervention is projected to avert roughly 1.05 million LRTI deaths (29% of all LRTI

**Table 2** Summary of disease burden, impact and cost-effectiveness ratios, with and without intervention (2030–2039), baseline scenario

| Country group by | N | Disease burden without intervention | | | | Burden averted and ICER with RSV maternal vaccine | | | | | Burden averted and ICER with RSV mAb | | | | |
|---|---|---|---|---|---|---|---|---|---|---|---|---|---|---|---|
| | | Non-severe cases | Severe cases | Hospitalisations | Deaths | Non-severe cases | Severe cases | Hospitalisations | Deaths | ICER per DALY averted | Non-severe cases | Severe cases | Hospitalisations | Deaths | ICER per DALY averted |
| **Gavi status** | | | | | | | | | | | | | | | |
| Gavi | 73 | 31288677 | 10683106 | 8031827 | 352990 | 2159630 | 1730164 | 1333545 | 83024 | 1073 | 13866799 | 4742022 | 3565171 | 182800 | 315 |
| Non-Gavi | 58 | 10657947 | 4599391 | 3457938 | 151973 | 819749 | 907088 | 699149 | 43528 | 1681 | 5610598 | 2441921 | 1835898 | 94133 | 577 |
| **World Bank income group** | | | | | | | | | | | | | | | |
| LIC | 34 | 10823869 | 3562172 | 2678130 | 117701 | 760348 | 577452 | 445078 | 27710 | 949 | 4774083 | 1573199 | 1182771 | 60645 | 257 |
| LMIC | 46 | 22502889 | 7872029 | 5918389 | 260107 | 1559549 | 1292990 | 996588 | 62046 | 1311 | 10083892 | 3536660 | 2658950 | 136334 | 428 |
| UMIC | 51 | 8619867 | 3848296 | 2893246 | 127155 | 659482 | 766809 | 591027 | 36796 | 1631 | 4619422 | 2074084 | 1559348 | 79954 | 551 |
| **WHO geographic region** | | | | | | | | | | | | | | | |
| EAP | 20 | 5313235 | 3097972 | 2329133 | 102363 | 314036 | 582849 | 449238 | 27969 | 1411 | 2570416 | 1558912 | 1172029 | 60094 | 479 |
| ECA | 20 | 2609512 | 633363 | 476178 | 20928 | 244307 | 125417 | 96666 | 6018 | 1425 | 1398536 | 337329 | 253612 | 13004 | 437 |
| LAC | 23 | 2583464 | 1024279 | 770079 | 33844 | 209296 | 204837 | 157881 | 9829 | 1507 | 1349631 | 536105 | 403057 | 20666 | 507 |
| MENA | 13 | 2907732 | 1058521 | 795823 | 34976 | 222211 | 192789 | 148594 | 9251 | 1566 | 1468837 | 535629 | 402699 | 20648 | 532 |
| SA | 8 | 12207891 | 4018853 | 3021474 | 132791 | 842810 | 643028 | 495622 | 30857 | 1138 | 5628427 | 1857416 | 1396452 | 71601 | 342 |
| SSA | 47 | 16324790 | 5449509 | 4097078 | 180062 | 1146719 | 888332 | 684693 | 42628 | 1169 | 7061551 | 2358554 | 1773220 | 90920 | 359 |
| Total | 131 | 41946624 | 15282497 | 11489765 | 504963 | 2979379 | 2637252 | 2032693 | 126552 | 1342 | 19477397 | 7183943 | 5401069 | 276933 | 431 |

DALY, disability-adjusted life year; EAP, East Asia & Pacific; ECA, Europe & Central Asia; ICER, incremental cost-effectiveness ratio; LAC, Latin America & Caribbean; LIC, low-income country; LMIC, low-income and middle-income country; MENA, Middle East & North Africa; RSV, respiratory syncytial virus; SA, South Asia; SSA, Sub-Saharan Africa; UMIC, upper-middle-income country.

deaths) among children younger than 6 months of age in LMICs.

## Cost-effectiveness of interventions

The average annual cost of vaccination programmes across all countries for the duration of analysis was estimated to be about US$546.36 million and US$538.40 million for RSV maternal vaccine and mAbs, respectively. The economic benefits expressed in terms of cost-of-care averted was about US$602.10 million (maternal vaccine) and US$1.97 billion (mAbs) over the 10 years (see online supplemental table 1).

For maternal RSV vaccine, the ICER per DALY averted is estimated at US$1342 (US$1073 across Gavi-eligible countries and US$1681 across non-Gavi countries). Similarly, the ICER estimates for RSV mAbs is US$431 (US$315 across Gavi-eligible countries and US$577 across non-Gavi countries). It is important to note these ICERs reflect the full potential cost of either intervention. Countries eligible for Gavi support would be expected to pay a share of the prices used in this analysis, thus reducing the ICER from the country perspective.

Results from alternative scenarios with low and high efficacy and duration of protection assumptions show that costs per DALY averted across countries range from US$244–US$1982 (maternal vaccine) and US$239–US$1958 (mAbs). By reducing the intervention price to 50% of the baseline price (ie, US$1.50 for Gavi-eligible countries and US$2.50 for non-Gavi countries), the average ICER per DALY averted would decline to US$781 (range US$45–US$1147) for the maternal vaccine and US$178 (range US$42–US$1132) for the mAb. Increasing the intervention price by 200% of the baseline price, the average ICER per DALY averted increases to US$2465 (range US$642–US$3651) for the maternal vaccine and US$938 (range US$632–US$3610) for the mAbs.

When comparing ICERs against an individual country's income level at baseline, the maternal vaccine ICERs were <50% of the GDP per capita in 60 countries (12 Gavi and 48 non-Gavi), suggesting intervention cost-effectiveness in those countries. ICERs for RSV mAbs were below the 50% GDP per capita threshold in 118 countries (62 Gavi-eligible and all non-Gavi). For both interventions, countries with higher ICER to GDP per capita ratios are concentrated in sub-Saharan Africa and Asia (figures 1 and 2). Many of these countries remain eligible for Gavi support and are expected to pay lower intervention prices. As a result, the cost per DALY averted from the perspective of these countries is likely to be much more favourable than shown here. For example, if each of the original Gavi-eligible countries were responsible for half of the cost of the intervention (US$1.50), which is still a relatively high cost as the countries with the lowest GDP per capita would pay only a fraction of that price under Gavi's current cofinancing model, then the ICER for the RSV maternal vaccine and mAb would fall below the 50% GDP per capita threshold in 46% (maternal vaccine) and 100% (mAb) of these countries. Further,

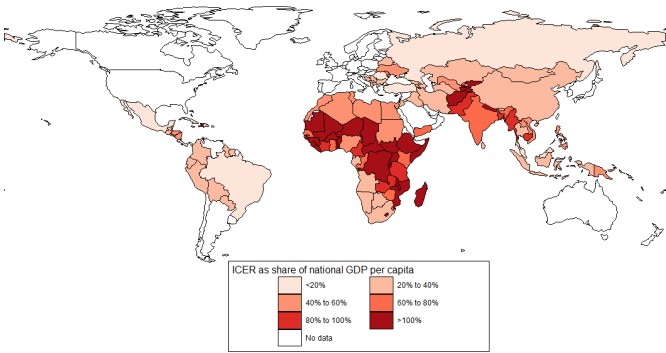

**Figure 1** ICERs as a percentage of national GDP per capita, maternal vaccine. GDP, gross domestic product; ICERs, incremental cost-effectiveness ratios.

maternal vaccine ICERs across countries at base price are roughly equivalent to the mAb ICER evaluated at 300% of the base price. Online supplemental table 2 includes a comparison of ICERs as a share of country GDP for alternative intervention scenarios.

## DISCUSSION

Both RSV interventions are projected to be impactful across all countries under baseline assumptions. A maternal vaccine is projected to avert 12 650 deaths and mAbs roughly two times more (27 690 deaths averted) annually among children younger than 6 months of age. We note that our baseline assumptions for the maternal vaccine draw from a phase III trial in which the primary endpoints were not met. As a result, maternal vaccine assumptions may be conservative compared with mAb assumptions, leading to lower overall impact of RSV maternal vaccines. Under alternative scenarios that consider both interventions with similar characteristics, we observe no substantial variation in impact. Under a minimal (30% efficacy and 4 months protection) and maximal (90% efficacy and 6 months protection) intervention characteristics scenario, both interventions are projected to avert roughly 84 900 and 356 000 deaths among children younger than 6 months of age across 131 countries, suggesting that efficacy and duration

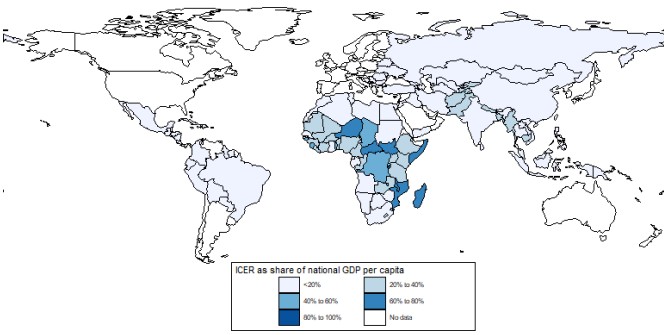

**Figure 2** ICERs as a percentage of national GDP per capita, monoclonal antibody. GDP, gross domestic product; ICERs, incremental cost-effectiveness ratios.

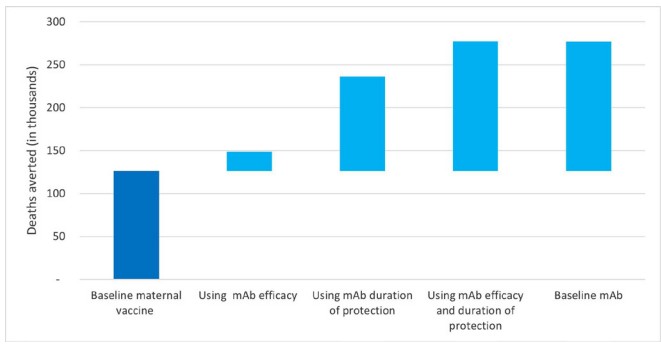

**Figure 3** Impact of change in key input parameter values on deaths averted. mAb, monoclonal antibody.

of protection are primary parameters for determining impact, reinforced by a similar study.[27]

Unknowns around intervention delivery strategy and potential coverage implications create uncertainties; this is especially true for a novel intervention like a maternal vaccine. To further understand the potential implications of unknown parameters on a maternal vaccine impact, we evaluated the marginal gains in impact by incrementally changing the parameter values to mimic those used in the mAb baseline scenario. When changing maternal vaccine coverage assumptions to the mAb coverage values, the maternal vaccine would prevent 22 000 additional deaths. Similarly, when changing both duration of protection and efficacy for maternal vaccine at baseline to the mAb baseline equivalent, maternal vaccine would avert an additional 150 000 deaths. As seen in figure 3, the duration of protection is the most important factor for increasing impact (109 000 additional deaths averted).

The cost per DALY averted under the baseline scenario for a maternal vaccine is more than three times that for mAbs (US$1342 vs US$431). This is mainly driven by the modest vaccine efficacy and assumed duration of protection for the maternal vaccine as compared with mAbs. Under the maximum and minimum scenarios with high and low vaccine efficacy and duration of protection

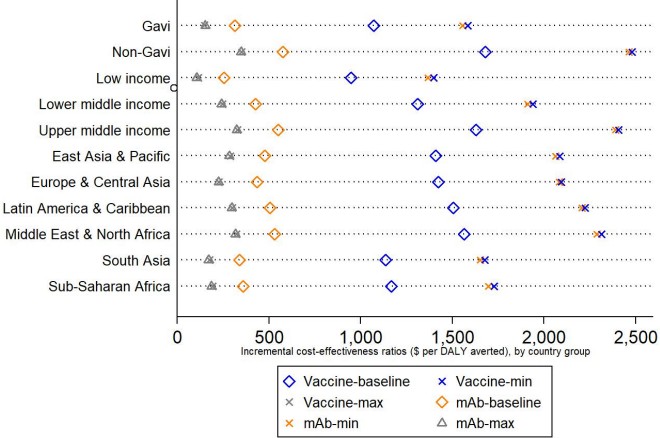

**Figure 4** Average incremental cost-effectiveness ratios by country groups. DALY, disability-adjusted life year; mAb, monoclonal antibody.

assumptions, the difference in the estimated ICERs between the two interventions is muted (figure 4).

Though it did not meet the primary endpoint, the recent phase III maternal vaccine trial shows promising impact on all-cause LRTI mortality.[11] If both future RSV interventions reduce all-cause LRTI mortality, our adjunct scenario shows more pronounced impact by averting more than a million all-cause LRTIs during the 10-year period. ICER estimates under this scenario were US$896 for the maternal vaccine (range US$34–US$7602) and US$889 for the mAb (range US$33–US$7608) per DALY averted across all countries, with 116 countries (69 Gavi-eligible and 47 non-Gavi countries) demonstrating ICERs <50% of their respective GDP per capita. We refrained from directly comparing these estimates to other scenarios as they use data sources[12] not comparable with the primary disease burden data[2] used in other scenarios.

There are several additional limitations worth citing. There is a dearth of RSV disease burden data, especially regarding the age distribution of disease in young infants in LMICs. Although we used the best published estimates of RSV disease burden in children,[2] the literature is expanding rapidly. For example, studies from Zambia[40] and Argentina[41] highlight that community mortality and deaths from RSV could be as high as 10% and 11% of all-cause deaths occurring among infants up to 6 months of age. This highlights a large and underappreciated burden of RSV and would mean our estimates of impact and effectiveness are conservative. Although we attempted to quantify the potential benefits of RSV interventions with additional scenario analysis, lack of consistent input data coupled with poorly established age distribution limits the comparability of our results across these scenarios. Collecting more granular data on disease burden is critical to inform future studies.[27]

The products evaluated in this study are not yet available in the market, so other key parameters are unknown. We assumed the same price for both interventions, which may not hold, as historical evidence suggests mAbs are likely to be more expensive to produce than a vaccine.[5] This could have considerable impact on the ICERs and comparisons between products. Nonetheless, our analysis shows that the mAb is more cost-effective than a maternal vaccine at baseline efficacy and duration of protection values, until a mAb reaches approximately three times the baseline price assumption. Gavi evaluated both interventions for inclusion in its 2018 Vaccine Investment Strategy in anticipation of the potential benefits, and they are expected to be included in the Gavi portfolio, subject to licensure, prequalification and affordability. In that case, the eligible Gavi countries would benefit from a considerable subsidy for access and affordability, especially the countries with the lowest GDPs per capita. Further, the <50% of GDP per capita thresholds used in this paper are non-specific measures of cost-effectiveness, especially when intervention prices to be paid by individual Gavi-supported countries are not yet known. Country-specific thresholds are recommended[42] but often do not exist for

most LMICs. In the absence of country-specific thresholds, we used a conservative metric uniform across all countries to define cost-effectiveness.

Lastly, RSV infection is seasonal in many countries. We did not consider seasonal delivery in this analysis. Seasonal intervention could potentially be a more cost-effective yet feasible strategy,[26] especially when using mAbs to selectively immunise children before the start of the RSV season. Delivering maternal vaccine seasonally to pregnant woman in LMICs may be more challenging due to the lack of a defined maternal vaccine delivery strategy. Future research should explore the feasibility of alternative delivery strategies.

## CONCLUSIONS

RSV interventions evaluated in this study are projected to be impactful and cost-effective across many LMICs. Under the assumptions used, mAbs are comparatively more impactful and cost-effective than RSV maternal vaccines. However, we reiterate the uncertainty around several critical parameters that inform this finding. The emerging evidence of RSV's role in all LRTI deaths among young infants suggests our analyses of RSV burden averted may prove conservative and enhance the attractiveness of RSV interventions as important tools for curbing LRTI mortality in infants. As disease burden shifts toward neonates and very young children, RSV maternal immunisation and mAbs offer the opportunity to protect young infants from disease. As RSV interventions complete clinical development and the intervention characteristics and market prices becomes more definitive, future analysis will provide additional clarity on the anticipated health and economic impacts of these interventions.

**Acknowledgements** We would like to thank Susan Nazarro for her critical feedback on this manuscript.

**Contributors** CP, DH and RB conceptualised the study. RB and CP developed the model. RB performed the analysis. RB and CP wrote the first draft of the paper. DH and KR reviewed and edited the manuscript.

**Funding** This work was funded by a grant from the Bill & Melinda Gates Foundation [OPP1088264], Seattle, Washington, USA.

**Disclaimer** The findings and conclusions contained within are those of the authors and do not necessarily reflect positions or policies of the Bill & Melinda Gates Foundation.

**ORCID iD**
Ranju Baral http://orcid.org/0000-0002-3043-6070

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
