## [Reviewer comments · BMJ Open]

ARTICLE DETAILS

TITLE (PROVISIONAL)	Impact and cost-effectiveness of potential interventions against infant respiratory syncytial virus (RSV) in 131 low- and middle-income countries using a static cohort model
AUTHORS	Baral, Ranju; Higgins, Deborah; Regan, Katie; Pecenka, Clint

VERSION 1 – REVIEW

REVIEWER	Sparrow, Erin World Health Organization, Immunization, Vaccines and Biologicals
REVIEW RETURNED	20-Jan-2021

GENERAL COMMENTS	This is a well written article and important for the RSV field as in the future LMICs will need to use CEAs to help make decisions about introduction of RSV maternal vaccines or mAbs in their countries. As with all modeling studies, assumptions are needed and these are clearly listed. However, I have some concern with using the Novavax trial results as the baseline assumption for maternal vaccine efficacy, but I acknowledge that in the absence of any other clinical data this is all we have and it is good to see that they have modelled a range of VE values out until 90%. It is stated elsewhere, but I think it is useful to repeat at line 133 that other maternal vaccines are in clinical development, which may have improved efficacy. In addition, I think that assuming 5 months duration for the mAb is better and I note below that there is an issue with how the WHO PPC has been interpreted. In addition the MEDI8897 product, which will likely be the first to market, is only looking out to 150 days. Below are some specific comments that should be addressed. Abstract • Line 20 - there is a typo, delete “for”• As the price assumptions are listed upfront in the abstract it would be useful to list also the efficacy and duration of protection baseline assumptions here. This would be useful for those readers who only look at the abstract. Introduction • Line 63 to 65 – “young children” should be changed to “infants” in the following sentence. “Multiple maternal vaccine candidates designed to protect against RSV illness in young children are in relatively advanced stages of development and expected to be available for global use in the coming years”• Line 65-67 - the authors should add that the current mAb product is not only expensive but that it also requires monthly dosing and add that long acting (not only more affordable mAbs) are in development which would be easier to deliver in LMIC settings.
---

	Methods  • Line 130. While I agree with the gestational age assumption which is likely to optimize transfer of maternal antibodies, please note that the WHO PPC for maternal vaccines does not specify an age window for maternal vaccination, it simply says women in the second or third trimester of pregnancy. https://apps.who.int/iris/bitstream/handle/10665/258705/WHO-IVB-17.11-eng.pdf;jsessionid=4D51C5B559A1D1C1F12EF913E99BAED5?sequence=1. This should be corrected or another reference provided. A possible reference could be the clinical trial registration of the Pfizer product that shows this as the age range in the phase 3 trial: https://clinicaltrials.gov/ct2/show/NCT04424316 • Line 136. The draft WHO PPC for RSV mAbs actually sets an efficacy target of 5 months (not 6). This was likely confused with the target population parameter stating through 6 months of age, but this is in reference to infants from birth to 6 months of age receiving a dose of mAb, rather than duration of protection. See: https://apps.who.int/iris/bitstream/handle/10665/258705/WHO-IVB-17.11-eng.pdf. It is not a problem to state 6 months of protection as an assumption but the WHO PPC is not the correct reference, otherwise I would suggest it be changed to 5 months which would also bring it in line with the lead candidate target of 150 days. • Line 141. The price assumption for the MAb product seems a little low. Perhaps a note should be added here to say that mAbs are likely to be more expensive to produce than vaccines. Based on current CHO cell production 1gram of mAb costs about \$100 to produce, so cost of good alone would be \$5 to produce a 50mg dose, never mind all the added markup. I appreciate that different costs are modelled and reported on lines 224 to 229 and that this is noted in the discussion in lines 298 to 300 but useful to re-iterate around line 141. Conclusion:  • It would be useful to add a forward looking statement. Something along the lines of as “As products complete clinical development and efficacy and duration of protection as well as market price become known this model should be repeated”
--	---

REVIEWER	Nenna, Raffaella Sapienza University of Rome
REVIEW RETURNED	17-Feb-2021

GENERAL COMMENTS	The manuscript by title “Impact and cost-effectiveness of potential interventions against infant respiratory syncytial virus (RSV) in low- and middle-income countries” is an interesting paper, analyzing the potential impact and cost-effectiveness of maternal vaccines and mAb therapy against RSV in infants across 131 countries. RSV infection in infants represents a leading cause of hospitalization both in developed and low-income countries. The topic covered by the Authors is up-to-date and many implications derive from the health cost analysis. The manuscript is clear and well-written, discussion and conclusions are supported by results. The revision significantly improved the quality of the manuscript.
--

VERSION 1 – AUTHOR RESPONSE

Reviewer: 1

Comments to the Author:

This is a well written article and important for the RSV field as in the future LMICs will need to use CEAs to help make decisions about introduction of RSV maternal vaccines or mAbs in their countries.

As with all modeling studies, assumptions are needed and these are clearly listed. However, I have some concern with using the Novavax trial results as the baseline assumption for maternal vaccine efficacy, but I acknowledge that in the absence of any other clinical data this is all we have and it is good to see that they have modelled a range of VE values out until 90%. It is stated elsewhere, but I think it is useful to repeat at line 133 that other maternal vaccines are in clinical development, which may have improved efficacy.

Author's response: Thank you for reviewing this manuscript and for the positive feedback. As suggested, we added a statement "other maternal vaccines are in clinical development which may have improved efficacy" in describing the intervention characteristics (line 133).

In addition, I think that assuming 5 months duration for the mAb is better and I note below that there is an issue with how the WHO PPC has been interpreted. In addition the MEDI8897 product, which will likely be the first to market, is only looking out to 150 days.

Author's response: Thank you for the feedback. Given the immunization products are still under clinical development, there is uncertainty around their final characteristics. As you very well noted, we have iterated this point throughout the manuscript. The 5 month protection falls within the 4-6 months range analyzed in the paper.

Below are some specific comments that should be addressed.

Abstract

- Line 20 - there is a typo, delete "for"

Author's response: Thanks for catching this. Deleted "for" from line 20.

- As the price assumptions are listed upfront in the abstract it would be useful to list also the efficacy and duration of protection baseline assumptions here. This would be useful for those readers who only look at the abstract.

Author's response: Thanks for the feedback. We added the vaccine efficacy and duration of protection values in abstract.

"The assumed baseline efficacy and duration of protection were higher for the mAb (60-70% efficacy, protection 6 months) compared to the maternal vaccine (40-60% efficacy, protection 3 months)."

Introduction

- Line 63 to 65 – "young children" should be changed to "infants" in the following sentence. "Multiple maternal vaccine candidates designed to protect against RSV illness in young children are in relatively advanced stages of development and expected to be available for global use in the coming years"

Author's response: Done.

- Line 65-67 - the authors should add that the current mAb product is not only expensive but that it also requires monthly dosing and add that long acting (not only more affordable mAbs) are in development which would be easier to deliver in LMIC settings.

Author's response: Thanks for the suggestion. We revised the original statement "Monoclonal

antibodies (mAbs) are available and in use for high-risk babies in high-income countries, but more affordable mAbs are also in advanced stages of development” to read

“ Monoclonal antibodies (mAbs) are available and in use for high-risk babies in high-income countries. However, the available mAbs are not only expensive but require multiple doses during the RSV season. Long lasting more affordable mAbs that are easier to deliver in low resource setting are in advanced stages of development.”

Methods

- Line 130. While I agree with the gestational age assumption which is likely to optimize transfer of maternal antibodies, please note that the WHO PPC for maternal vaccines does not specify an age window for maternal vaccination, it simply says women in the second or third trimester of pregnancy. <https://apps.who.int/iris/bitstream/handle/10665/258705/WHO-IVB-17.11-eng.pdf;jsessionid=4D51C5B559A1D1C1F12EF913E99BAED5?sequence=1>. This should be corrected or another reference provided. A possible reference could be the clinical trial registration of the Pfizer product that shows this as the age range in the phase 3 trial: <https://clinicaltrials.gov/ct2/show/NCT04424316>

Author’s response: We agree that the WHO PPC does not specify 24-36 weeks explicitly. We added the term inferred to clarify this point. Thanks for sharing the additional reference. We revised the sentence and added this additional reference to accommodate the concern. The revision reads:

“Our analysis assumed a single dose maternal RSV vaccine would be given to pregnant women between 24 and 36 months of gestation, inferred based on the WHO preferred product characteristics (PPCs) [22] and other ongoing clinical trials [23].”

- Line 136. The draft WHO PPC for RSV mAbs actually sets an efficacy target of 5 months (not 6). This was likely confused with the target population parameter stating through 6 months of age, but this is in reference to infants from birth to 6 months of age receiving a dose of mAb, rather than duration of protection. See: <https://apps.who.int/iris/bitstream/handle/10665/258705/WHO-IVB-17.11-eng.pdf>. It is not a problem to state 6 months of protection as an assumption but the WHO PPC is not the correct reference, otherwise I would suggest it be changed to 5 months which would also bring it in line with the lead candidate target of 150 days.

Author’s response: Thanks for clarifying this point. We added the term “inferred” based on the WHO PPC to clarify this point. In addition, we also added other references.

“Our analysis assumed a single dose mAb would be given to newborns at birth, would have 60% to 70% efficacy, and would offer protection for six months, inferred based on the PPCs [23] and other studies [Cromer et al and Li et al].”

- Line 141. The price assumption for the MAb product seems a little low. Perhaps a note should be added here to say that mAbs are likely to be more expensive to produce than vaccines. Based on current CHO cell production 1gram of mAb costs about \$100 to produce, so cost of good alone would be \$5 to produce a 50mg dose, never mind all the added markup. I appreciate that different costs are modelled and reported on lines 224 to 229 and that this is noted in the discussion in lines 298 to 300 but useful to re-iterate around line 141.

Author’s response: We completely agree with your point that the assumed mAb price is lower than expected for the reasons you highlighted. We added a sentence in the manuscript (changes underlined here)

“For both interventions, we assumed a per-dose price of US\$3 in Gavi-eligible countries and per-dose price of \$5 in LMICs not eligible for Gavi support. Traditionally mAbs are more expensive to produce than a vaccine and will likely have much higher market price than a vaccine. If Gavi decides

to support RSV interventions once they are available, Gavi-eligible countries would likely be able to access the interventions at varying prices depending on their transition status [24]. We refrained from projecting individual country Gavi eligibility or intervention prices due to significant uncertainty, and instead evaluated a range of intervention prices in sensitivity analyses.”

Conclusion:

- It would be useful to add a forward looking statement. Something along the lines of as “As products complete clinical development and efficacy and duration of protection as well as market price become known this model should be repeated”

Author’s response: Thanks for the feedback. We added the following sentence (underlined in the conclusion section.

“RSV interventions evaluated in this study are projected to be impactful and cost-effective across many LMICs. Under the assumptions used, mAbs are comparatively more impactful and cost-effective than RSV maternal vaccines. However, we reiterate the uncertainty around several critical parameters that inform this finding. The emerging evidence of RSV’s role in all LRTI deaths among young infants suggests our analyses of RSV burden averted may prove conservative and enhance the attractiveness of RSV interventions as important tools for curbing LRTI mortality in infants. As disease burden shifts toward neonates and very young children, RSV maternal immunization and mAbs offer the opportunity to protect young infants from disease. As RSV interventions complete clinical development and the intervention characteristics and market prices becomes more definitive, future analyses will provide additional clarity on the anticipated health and economic impacts of these interventions.”

Reviewer: 2

Comments to the Author:

The manuscript by title “Impact and cost-effectiveness of potential interventions against infant respiratory syncytial virus (RSV) in low- and middle-income countries” is an interesting paper, analyzing the potential impact and cost-effectiveness of maternal vaccines and mAb therapy against RSV in infants across 131 countries.

RSV infection in infants represents a leading cause of hospitalization both in developed and low-income countries. The topic covered by the Authors is up-to-date and many implications derive from the health cost analysis. The manuscript is clear and well-written, discussion and conclusions are supported by results. The revision significantly improved the quality of the manuscript.

Author’s response: Thank you for your time in reviewing our manuscript. We appreciate your positive feedback.

VERSION 2 – REVIEW

REVIEWER	Sparrow, Erin World Health Organization, Immunization, Vaccines and Biologicals
REVIEW RETURNED	06-Apr-2021

GENERAL COMMENTS	The revision of this manuscript looks good for publication and I have not further comments. This is an important paper for the field.
---